# New Insights in the Diagnostic Potential of Sex Hormone-Binding Globulin (SHBG)—Clinical Approach

**DOI:** 10.3390/biomedicines13051207

**Published:** 2025-05-15

**Authors:** Weronika Szybiak-Skora, Wojciech Cyna, Katarzyna Lacka

**Affiliations:** 1University Clinical Hospital in Poznan, Poznan University of Medical Sciences, 60-355 Poznan, Poland; weronikaszybiak@gmail.com; 2Student’s Scientific Society, Poznan University of Medical Sciences, 60-355 Poznan, Poland; cynawojtek2@gmail.com; 3Department of Endocrinology, Metabolism and Internal Medicine, Poznan University of Medical Sciences, 60-355 Poznan, Poland

**Keywords:** SHBG, metabolic syndrome, cardiovascular, diabetes mellitus type 2, PCOS, cancer

## Abstract

SHBG is a glycoprotein that not only controls serum sex hormone levels but is also strongly correlated with metabolic syndrome, cardiovascular risk, thyroid function, gynecological conditions, and even the process of carcinogenesis. Synthesis of SHBG is controlled by many factors related to obesity, lipogenesis, inflammatory status, and genetic predisposition. By influencing the bioavailability of sex hormones, SHBG regulates their effects not only on the reproductive system, but also cardiomyocytes, vascular epithelium, and more. In this review, we aim to gather and summarize current knowledge on the physiology of SHBG and its association with cardiovascular disease, metabolic syndrome, DM 2, thyroid function, PCOS, hypogonadism, infertility, and its correlations with oral contraception. What is more, genetic alterations are mentioned to highlight SHBG as a potential new diagnostic marker. Furthermore, we assess the clinical usefulness of this parameter in the diagnosis and treatment of patients suffering from the above-specified conditions.

## 1. Introduction

Circulating hormones can be divided into two fractions: bound and free. The free fraction constitutes a small percentage of the total concentration of hormones in the bloodstream [1]. The remaining hormonal fraction is bound to plasma hormone-binding proteins, crucial in regulating the concentration of biologically active unbound hormones and their diffusion into cells [1].

Interplays between hormones and proteins are very complex and not yet fully understood. The subject of much research is Sex Hormone Binding Globulin (SHBG), which controls serum levels of sex hormones and their bioavailability [2].

This glycoprotein is produced mainly in hepatic cells, and the production process is dependent on many factors. Decreased synthesis of SHBG could be caused in course of metabolic disturbances [3,4,5]. SHBG level is negatively correlated with body mass index (BMI) status. Overweight and obese patients tend to present reduced SHBG levels, which normalize with weight loss [6]. Eating disorders such as anorexia nervosa, in turn, tend to increase SHBG levels, which may also affect hormonal and menstrual disorders in this group of patients [7,8].

What is more, cytokine and inflammatory responses influence the production of SHBG. The inflammation process promotes macrophages to produce cytokines, which downregulate SHBG gene transcription in the liver [9]. TNF-α itself, which may be increased in obese patients, leads to lower SHBG synthesis [10]. Inflammatory response and cytokines are also correlated with the excess of fat tissue and affect the intravascular epithelial cells, leading to an increased cardiovascular risk.

Metabolic abnormalities and alterations in SHBG levels contribute to an imbalance in circulating sex hormone levels, thereby impacting the reproductive system and associated disorders such as polycystic ovary syndrome (PCOS) [11], breast cancer [12], or infertility [13]. Interestingly, some studies confirmed that decreased levels of SHBG in women before pregnancy were found to be at an increased risk of developing metabolic disorders during gestation, like gestational diabetes mellitus or cardiovascular diseases, which increased the risk of miscarriage, developmental abnormalities in the fetus, and increased the risk of abnormal delivery [14,15].

Because of the significant correlation between SHBG and metabolic disorders, decreased levels of SHBG as a laboratory marker of increased risk of metabolic syndrome (MS) are being discussed, especially in the male population [16]. Some studies have confirmed that assessment of SHBG serum level could be a predictor for MS and risk for the development of MS’s phenotype [16]. Similarly, a lower level of SHBG is correlated with increased risk of diabetes mellitus type 2 (DM 2) development [17]. SHBG is also used to assess cardiovascular risk. Patients with numerous cardiovascular risk factors presented decreased SHBG serum levels [18].

In this study, we aim to collect and summarize current knowledge about the physiology of SHBG and its association with cardiovascular disease, metabolic syndrome, DM 2, thyroid function, PCOS, and infertility. Moreover, we want to evaluate the clinical usefulness of this parameter in the diagnosis and treatment of patients suffering from the above-mentioned conditions.

## 2. SHBG Synthesis

SHBG is mainly produced and secreted by the liver and in smaller quantities in the testis, duodenum, small intestine, and kidneys [19,20].

### 2.1. SHBG Gene

This glycoprotein is encoded by the *SHBG* gene located on the short arm of chromosome 17 (17p13.1) [20] and consists of 11 kb [21]. At least two transcripts are produced from this gene: a 4.3 kb transcript, which is expressed and secreted by hepatocytes, and a second transcript of 8 kb expressed in the testis [21]. SHBG produced in the germ cells of the testes, accumulates in the sperm acrosomes and may affect its function [22] through the inter-reference in the capacitation reaction [21].

The SHBG gene consists of eight exons. The first exon has three variants, 1L, 1T, and 1N, which are triggered by three promoters: the downstream promoter (PL), the upstream promoter (PT), and third, the upstream promoter (PN) [23]. SHBG is built of 1L, 2, 3, 4, 5, 6, 7, and 8 exons, which are connected. Moreover, we can also distinguish a variation: SHBG-T, which is a missing exon seven, but exon 1T is promoted by the promoter PT, which is included [23]. The location and structure of the gene as well as the structure of SHBG are schematically shown in Figure 1.

### 2.2. Transcriptional Factors

The expression of the gene for SHBG in liver cells is subject to complex regulatory processes. Weight loss, fasting, hyperthyroidism, or thyroid hormone treatment, growth hormone pulsatility promote hepatocyte nuclear factor 4 alpha (HNF-4α) activity, resulting in increased hepatic synthesis of the major SHBG fraction [24].

In contrast, overweight, obesity, increased lipogenesis, and hepatic steatosis exacerbate SHBG synthesis inhibitory factors (e.g., chicken ovalbumin upstream promoter transcription factor (COUP-TF) or peroxisome proliferator-activated receptor gamma (PPARγ)) [24].

Furthermore, it is important to remember that factors such as age, comorbidities, and environmental factors also affect genetic transcription, which is associated with changes in SHBG levels. Factors that influence SHBG synthesis are listed in Table 1.

Transcription of the gene is stimulated by HNF-4α, Erα, while inhibition of transcription occurs with COUP-TF and PPARγ [25]. Jänne et al. examined the proximal promoter of SHBG in the human HepG2 cell line.

It has been shown that the gene for SHBG has a proximal promoter without TATA and has two trace regions (FP1 and FP3) binding COUP-TF and HNF-4α, which compete with each other for binding sites [25].

Lopez et al. presented that constitutive androstane receptor (CAR), which is involved in detoxification gene expression, increases SHBG production. CAR binds to a typical direct repeat 1 nuclear hormone receptor-binding element in the human SHBG proximal promoter [26]. Moreover, resveratrol contained in red wine, through the interaction with CAR, can increase SHBG serum level independently of alterations in HNF-4α or PPARγ levels [26].

PPARγ presents affinity to the proximal promoter of the SHBG gene. The interaction of PPARγ with the promoter leads to a decrease in gene expression and hepatic SHBG production [27].

The expression of transcription factors, and thus their effect on SHBG synthesis, also depend on cytokine profile (tumor necrosis factor alpha (TNF-α) or interleukin-1β (IL-1 β) [10,27,28]. Simó et al. investigated that increased TNF-α in obese patients can cause decreased serum levels of SHBG. The process of downregulation of SHBG is indirectly mediated by TNF-α, which inhibits HNF-4α gene expression through nuclear factor-kappa B (NF-κβ) [10]. NF-κβ is a nuclear factor that can bind to DNA; therefore, it can regulate gene transcription, playing a role in immune processes, autoimmune inflammation, apoptosis, and other cell processes [29]. The inhibition of NF-κβ resulted in increased production of SHBG even in the presence of TNF-α [10].

Moreover, de novo lipogenesis causes a decrease in HNF-4α expression. Ruth et al. showed that the variability of proteins involved in the process of triglyceride synthesis from carbohydrates influences the level of SHBG [30,31]. Gene structure and transcription, alongside factors influencing this process, are presented in Figure 2.

It is worth mentioning that the HepG2 cells on which the SHBG gene transcription process was studied belong to the tumor cell group and are derived from hepatoblastoma, which may affect the transcriptional process. Cells of the HepG2 line present a mutation of the *CTNNB1* gene and an increased number of chromosomes. The main difference between the HepG2 line and normal hepatocytes is the much lower or absent expression of the cytochrome P450 (CYP) superfamily, which may affect oxidation processes. Nevertheless, this cell line has most of the metabolic properties of normal hepatocytes and is therefore often used during in vitro studies [32].

## 3. Clinical Usefulness of SHBG

SHBG transports steroid sex hormones and regulates their bioavailability in a variety of tissues. SHBG is also a component of many processes involved in metabolism at the cellular level. Its activity affects not only the reproductive system, but also adipose tissue, hepatocytes, and cardiomyocytes. SHBG may represent a crucial link between imbalances in hormone levels and many disease entities. Disease entities in which SHBG may play an important role in the diagnostic and therapeutic process are described below. These conditions are listed in Table 2.

### 3.1. SHBG and Metabolic Syndrome

Over recent years, an increase in the incidence of MS has been noticed in various age groups, which is related to the generational neglect of healthy eating habits, reduction of physical exercise, and increased levels of stress [33].

According to the meta-analysis investigated by Brand et al., MS diagnosis is connected with disbalance in sex-hormone levels and SHBG. Both male and female patients with MS presented reduced levels of SHBG compared to the healthy population. Whereas levels of androgens—free testosterone levels and total testosterone levels, were decreased in the male group with MS. In contrast, women with MS presented increased levels of both free and total testosterone [34].

Patients with MS diagnosis present disbalance in serum lipid profile and increased circumference of the waist, which is a result of the increased fat tissue concentration [35]. Processes involved in lipid metabolism and de novo lipogenesis can also impact the SHBG production and function [36]. High carbohydrate intake, especially fructose and sucrose, stimulates the process of lipogenesis in the liver. Intake of sugar stimulates the expression of fatty acid synthase and acetyl-CoA carboxylase [37].

Selva et al. demonstrated that monosaccharides (glucose and fructose) reduce human SHBG production by hepatocytes in transgenic mice and HepG2 hepatoblastoma cells. The de novo lipogenesis process causes a decrease in HNF-4α expression and thus a decrease in SHBG levels; also, mRNA for SHBG was decreased [38]. Moreover, de novo lipogenesis promotes HNF-4α replacement by COUP-TF1, which is connected with repression of SHBG transcriptional activity. The process of lipogenesis leads to increased cellular levels of palmitate, which result in downregulation of HNF-4α and indirectly decreased SHBG [38].

Increased levels of both glucose and fructose resulted in decreased SHBG expression. Therefore, the question arises: What effect does insulin have on the SHBG synthesis process? Selva et al. proved that an increase in insulin levels does not affect SHBG levels. In cells in which the monosaccharide level was increased and the insulin level was lowered, the suppression of SHBG expression was maintained. These studies indicate that decreased insulin levels do not cause an increase in SHBG [38].

Similar results were obtained by Simons et al. In this study, a negative correlation between enhanced de novo lipogenesis and SHBG levels was also established, but especially in the female population. In the male group of patients, the correlation was not statistically significant [39]. Increased intake of carbohydrates, especially monosaccharides, and enhanced de novo lipogenesis can be the answer for the correlation between increased BMI and decreased SHBG serum level [40]. Decline in serum SHBG levels resulted in increased serum levels of active androgens and hyperandrogenism presentation in overweight and obese women [1]. In contrast to the population with increased BMI, patients with anorexia nervosa present elevated serum levels of SHBG [41], which could be connected with menstrual cycle and reproductive system disturbances.

Both SHBG levels and adiponectin present a negative correlation with BMI. Adiponectin is a peptide synthetized by white adipose tissue, regulating metabolic processes [42]. Adiponectin influences the insulin tissue sensitivity, regulates the level of glucose and lipid metabolism. In the liver, it acts through the activation of special receptors AdipoR2, which results in AMPK pathways activation and influence on the decreased process of de novo lipogenesis. Activation of AMPK pathways also increased oxidation of fatty acids in hepatic cells, which resulted in decreased levels of lipids in hepatocytes [43]. Simó et al. demonstrated that HepG2 cells treated with adiponectin increased SHBG levels. The influence of adiponectin on lipid metabolism increases HNF-4α, which enhances SHBG transcription [36]. This effect of adiponectin was confirmed in other studies [44,45], also in the child population [46]. In a previous study, adiponectin intake was correlated with decreased levels of HNF-4α, but in this research, shorter treatment duration and lower doses of adiponectin were used [47]. Correlation between the MS process and SHBG level is presented in Figure 3.

Overweight and obesity are directly correlated with enhanced inflammatory processes through the occurrence of TNF, TNF-α, or IL-1β [5,36]. Weight gain and obesity lead to phenotype changes in adipocytes. Inflammation-altered adipocytes are responsible for increased cytokine production. Cytokines produced by adipocytes promote leukocyte migration and fat tissue infiltration [48]. Simo et al. investigated that TNF-α can cause decreased serum levels of SHBG by an indirect inhibition of HNF-4α gene expression through NF-κβ. The inhibition of NF-κβ resulted in increased production of SHBG even in the presence of altered TNF-α levels [5]. It is also confirmed that IL-β decreases SHBG expression through the pathways connected with HNF4-α [9].

Moreover, Jarecki et al. established that assessment of SHBG can be a predictor of MS occurrence. They investigated that decreased SHBG plasma levels in combination with elevated IL-18 serum concentrations in men significantly increase the risk of MS [16].

The correlation between lowered levels of SHBG and inflammatory process indicates that not only metabolic disorders but also other pro-inflammatory diseases like osteoarthritis, bowel diseases, cancers, or allergies can have an influence on SHBG and serum levels of sex hormones [13].

### 3.2. SHBG and Cardiovascular Risk

Sex hormones seem to be an important factor for cardiovascular condition control; however, the pathophysiology of their interaction still remains unclear. It is considered that estrogens possess a cardioprotective effect in pre-menopausal women, and the risk of heart diseases increases after menopause [49]. In men, testosterone is a crucial sex hormone involved in maintaining cardiovascular condition [50]. Physiological levels of testosterone play a protective role against cardiac and vascular incidences, but decreased levels of testosterone were connected with elevated risk of a negative metabolic profile and an increase in the cardiovascular incidence ratio [49]. As is well known, normal concentrations of free androgens are maintained by adequate levels of SHBG. According to the previous study, levels of sex steroids influenced the cardiovascular condition [50]. SHBG serum levels might be one of the crucial markers in cardiovascular risk assessment.

SHBG acts not only as a transporter of hormones but also regulates their levels in individual tissues [51]. Schock et al. established that cardiac cells present expression of androgen-binding protein (ABP), which has a similar structure to SHBG. The above study evaluated myocardial biopsy material taken from a group of male patients with dilated cardiomyopathy. The results of this study indicate that ABP regulates the availability of androgen in the cardiac muscle [52].

Moreover, SHBG can independently interact with membrane receptors in some human tissues [53]. In human cardiac myocytes, plasma membrane receptors for SHBG have been identified [4,52]. LG domains of SHBG can bind membrane receptors, with tyrosine kinase activity and G-protein-coupled receptors. Interaction between SHBG and G-protein-coupled receptors leads to stimulation of the cAMP pathway [53]. The effects of SHBG on potential receptors on myocardial cells may represent one putative mechanism for the effects of SHBG on cardiomyocytes. cAMP has been shown to inhibit cardiac hypertrophy through peroxisome proliferator-activated receptor-α (PPARα) [54]. In vitro study using adult rat cardiomyocytes revealed that chronic activation of cAMP may lead to hypertrophy and heart failure. Compartmentalization of AMP leads to redistribution of protein kinase A and significantly reduces its activation with potential consequences for myocyte contractility [55].

Interestingly, SHBG can be internalized in steroid-dependent tissue [56]. Tissues that are under the influence of steroid hormones can internalize SHBG to activate signal transduction pathways independently of the classical mechanisms based on intracellular androgen receptors [56]. In light of the above correlations, it is worth emphasizing that studies using human cells are needed to establish the exact relationship between SHBG activity and cardiomyocytes.

Internalization of SHBG can occur through the low-density lipoprotein-associated protein 2 receptor, megalin, also responsible for transporting processes in mitochondria. The receptor for megalin can affect nuclear and mitochondrial function, and what is more, it may also influence some SHBG effects. The human promoter of megalin is controlled by metabolic factors. Megalin expression is decreased in cells of rats with MS symptoms. Nevertheless, this relationship needs more adequate research [4,57].

Testosterone activates AMPK pathways in cardiomyocytes and increases glucose uptake through the GLUT4. The same pathways are also involved in increased synthesis of SHBG in hepatocytes. What is more, estrogens influence SHGB expression and synthesis. Estrogens, through binding to ERα, upregulate the synthesis of SHBG protein and thus regulate androgen bioavailability [58]. A study performed by Akdis et al. has proven that decreased levels of estrogens in female patients and elevated free testosterone levels in males can be independently connected with arrhythmogenic right ventricular cardiomyopathy and induction of major arrhythmic cardiovascular events [59].

Normal range testosterone levels, controlled by SHBG, decrease the activation of cardiac fibroblasts, resulting in a lower risk of heart failure development [60]. The study performed by Yeap et al. proved that men with decreased levels of SHBG presented increased risk of coronary heart disease (CHT) [61], confirmed in the meta-analysis conducted by Li et al., which established that increased levels of SHBG were independently correlated with decreased risk of CHT among both male and female patients [3]. Direct correlation between SHBG and CHT risk indicates that SHBG can be an independent marker in the assessment of cardiovascular risk. Consistent with previous studies, reduced SHBG levels were also correlated with lower levels of high-density lipoprotein (HDL) and elevated levels of C-reactive protein [3]. The above changes show an association with an increased risk of CHT. Serum SHBG levels are also a marker of metabolic liver dysfunction, which has also been shown to increase cardiovascular risk. Increased hepatic de novo lipogenesis and activation of proteins involved in carbohydrate metabolism not only affect the cardiovascular system but also reduce SHBG production [38]. Moreover, meta-analysis conducted by Li et al. indicated a dose–response causal relationship between SHBG and risk of CHD in both men and women, indicating that SHBG is not only a promising marker, but may also be a modifiable cardiovascular risk factor.

Saltiki et al. confirmed the genetic association between SHBG and atherosclerosis in healthy women [62]. The SHBG promoter (TAAAA)n polymorphism is linked with early markers of atherosclerosis (impaired endothelium-dependent vasodilatation, increased carotid artery intima media thickness). The longer alleles of the promoter of SHBG a higher risk of previous markers being found in healthy women [62].

The probable mechanisms of interaction between SHBG and the cardiovascular system are summarized in Figure 4.

### 3.3. SHBG and Insulin Resistance, Diabetes Mellitus, and Gestational Diabetes Mellitus

SHBG serum level is strongly correlated with metabolic diseases. Decreased level of SHBG is one of the risk factors for DM2 development [63].

The study conducted by Soriguer et al. has proved that male patients with decreased levels of testosterone and lower levels of SHBG have an increased risk of DM2. In the female population, decreased levels of SHBG were related to the development of insulin resistance (IR) [63]. Higher levels of SHBG were observed in women compared to men, which may indicate the protective effect of SHBG against the development of diabetogenic changes [63]. Another study confirmed that in perimenopausal women, increased SHBG levels were a protective factor against DM2 [64]. The meta-analysis conducted by Ding et al. indicated that women with SHBG serum levels >60 nmol/L had an 80% lower risk of DM2 development than female patients with decreased SHBG. In the male population, SHBG serum level >28.3 nmol/L was associated with a 52% reduced risk of DM2 [65]. The mechanism of this protective function of SHBG is still unclear.

IR is defined as the failure of insulin to provide the proper glucose transport into the tissues, which results in the development of increased levels of glucose and insulin serum levels [66]. Inappropriate metabolic-mitochondrial function is one of the factors leading to IR and DM2 development. Mitochondrial reactive oxygen species (ROS) and nitric oxide (NO) attenuate insulin activity in adipocytes and inhibit glucose intake through the GLUT4 [67]. Decreased intake of glucose leads to altered glucose and insulin serum levels, which cause androgen secretion stimulation as a consequence of insulin action, as well as SHBG through increased levels of monosaccharides [68]. As studies revealed, SHBG can influence mitochondrial activity. In the study conducted by Marycz et al., decreased levels of PPAR-γ lead to reduced cell viability, premature senescence, and profound mitochondrial failure in equine ASCs. Interestingly, SHBG supplementation improves mitochondrial dynamics and metabolism of cells. This study indicates that SHBG might serve as a factor for metabolic process modulation in mitochondria and regulation of the energy production process on the molecular level [67]. It is necessary to conduct more studies to clarify the mechanisms of the SHBG effects and whether they could be related to a PPARγ pathway. It should be noted that the study by Marycz et al. uses equine ASCs, so studies using human ASC cells seem particularly important due to species-specific limitations.

SHBG serum level is controlled by a lot of factors. One of them is lipogenesis, which decreases SHBG transcription and SHBG protein synthesis in the liver. Previously mentioned study proved that altered insulin serum levels do not have an influence on decreased SHBG synthesis. Elevated monosaccharide concentration was correlated with suppression of SHBG expression. Increased levels of monosaccharides promote the de novo lipogenesis process, which contributes downregulation of SHBG transcriptional factors [38]. Some studies confirmed that in the overweight population, increased visceral adipose tissue and increased stage of hepatic steatosis were connected with decreased SHBG production, which leads to increased risk of DM2 development [69]. On the other hand, patients with decreased SHBG levels are more susceptible to fat accumulation in the liver. This fact is suggestive that SHBG could be a factor controlling lipogenesis and fat tissue acquisition [70]. Body mass reduction seems to be crucial to decline insulin and glucose serum levels and induce increased insulin sensitivity of tissues. Bariatric surgery might be a therapeutic option, providing durable weight-loss connected with an increase in both testosterone and SHBG serum levels, which was also correlated with potentially sustained remission of DM2 [71].

Risk of DM2 development can also be correlated with specific SHBG genetic variants. Perry et al. analyzed and summarized 15 studies with a total number of 27,657 patients with DM2 and 58,481 controls. Based on the pooled analysis, the reference sequence (rs) 1799941 allele, which is associated with increased SHBG production, was associated with a lower risk of DM2 development in both male and female groups [72]. The rs6257 SHBG variant is associated with increased risk of DM2 due to the effect of lowering the concentration of SHBG in carriers. What is more, the rs6259 grants a protective effect against the risk of DM2, thus it increases the SHBG levels [73]. This phenomenon grants a protective effect against gestational diabetes mellitus as well, specifically the variant allele A that is found more commonly in healthy Northeast Chinese women in comparison to healthy African American and Caucasian women [74].

Some studies also indicate that SHBG can be an important factor involved in IR and glucose transport in gestational diabetes mellitus (GDM). GDM glucose intolerance that is first diagnosed during pregnancy is correlated not only with overweight or incorrect eating habits, but also with increased serum levels of hormones, which exacerbate IR [75]. Altered serum levels of sex hormones during pregnancy are transported and regulated by SHBG. A study conducted by Liu et al. investigated the influence of β-carotene on IR and glucose transport in GDM. It has been proven that the intake of β-carotene promotes SHBG and GLUT4 expression, indicating that β-carotene promotes glucose transport and inhibits IR in GDM by increasing the expression of SHBG [14]. A study conducted by Hedderson et al. also confirmed that low pre-pregnancy SHBG levels were associated with increased risk of GDM and might be useful in identifying women at risk for GDM for early prevention strategies [17]. Early-pregnancy screening for GDM conducted in the first trimester with the use of SHBG and adiponectin maternal serum concentration can increase detection of GDM to 74.1% [76]. In contrast, a study conducted by Yang et al. did not confirm the statistically significant importance of SHBG assessment in predicting GDM [77].

### 3.4. SHBG and PCOS, Hypogonadism and Infertility

#### 3.4.1. PCOS and SHBG

PCOS is the most common endocrine disorder in women of reproductive age. The prevalence of PCOS varies depending on the diagnostic method, geographical region, and diagnostic criteria [78]. The most popular diagnostic criteria, supported by the International Evidence-Based Guideline for the Assessment and Management of PCOS in 2018, are the Rotterdam criteria [78,79].

SHBG is not included in the Rotterdam diagnostic criteria. Nevertheless, SHBG plays a pivotal role in the management of sex hormones—testosterone and estradiol, affecting their bioavailability. An interesting study by Li et al. investigated polymorphisms of the *SHBG* gene, as the SHBG alterations may be associated with hyperandrogenism [11]. The meta-analysis revealed that the *SHBG* (TAAAA)n pentanucleotide repeats polymorphism (rs35785886) is a risk factor for PCOS and low SHBG serum concentration in PCOS [11]. In the study consisting of 1004 participants, SHBG mutant alleles of the rs727428 variant are responsible for a greater risk of PCOS and a lower concentration of SHBG in Mediterranean women [80]. Interestingly, the rs6259 variant was found to be a protective factor [80]. Although the association found in this study was weak, it nonetheless highlights the need for a more comprehensive understanding [80]. However, the rs6259 *SHBG* SNP GA + AA genotype has an impact on the in vitro fertilization–embryo transfer in PCOS patients that manifests by a lowered number of oocytes and embryos [81].

PCOS is connected with hyperandrogenism through binding and the bioactivity regulation of metabolic clearance rates for 5-dihydrotestosterone, testosterone, and 17-estradiol. The presence of SHBG receptors can be found on the surface of tissues like ovaries, endometrium, hypothalamus, and breasts, responsible for the nongenomic effect of sex hormones [82]. Thus, lower values of serum SHBG in this condition are expected. Indeed, meta-analysis conducted by Deswal et al. confirmed this susceptibility [82]. The SHBG concentration has clinical utility in predicting PCOS in patients [82]. Moreover, SHBG assessment may be useful to differentiate between PCOS and polycystic ovarian morphology (PCOM) with functional hypothalamic amenorrhea (FHA) [83]. Women with FHA are more likely to exhibit elevated levels of SHBG compared to those with PCOS. Accordingly, SHBG may serve as a useful biomarker in cases of diagnostic uncertainty regarding PCOS, particularly when differentiating from FHA-PCOM [83].

PCOS is responsible for many metabolic comorbidities, including obesity and IR [84,85]. Interestingly, lower SHBG levels in women with PCOS and co-occurring IR are found to be a positive factor for obesity [84,86]. This mechanism is explained by SHBG, which stands in a negative correlation with free androgen index (FAI) [84]. Interestingly, PCOS shows correlations between SHBG and adipokine concentrations. Patients with lower SHBG levels present higher leptin concentration and lower concentration of adiponectin, apelin-12, and apelin 36 [86].

#### 3.4.2. Infertility and SHBG

The prevalence of infertility in reproductive-age couples is estimated between 8% and 12% of the global population [87,88]. A considerable number of growing infertile couples around the globe encourages the research, thus the link between infertility and SHBG concentration is being widely discussed.

Recently, Ma et al. presented a Mendelian randomization study to assess the connection between female infertility and SHBG [13]. Interestingly, higher SHBG concentration in women provides a protective effect on female infertility, specifically in the case of anovulation. No relation to men’s and women’s infertility of tubal origin was revealed [13]. Furthermore, SHBG may contribute as a predictive value towards conception, pregnancy, and live births in this study group [89]. Women undergoing in vitro fertilization are found to be affected by SHBG levels and SHBG gene (TAAAA)n polymorphisms [90]. Lower SHBG levels are associated with large follicles, and shorter allele genotypes (TAAAA with less than eight repeats) result in a higher number of small follicles [90].

Another study performed in a group of men with primary infertility revealed a negative correlation between the SHBG concentration and BMI [91]. The role of SHBG in diagnosing primary male infertility can be a more precise factor to determine the accuracy of decreased semen parameters [92]. Genetic factors, such as *SHBG* gene alterations, affect male fertility, especially concerning rs6259 and rs727428 loci [93]. The rs6259 A allele is related to male infertility, in contrast to the rs727428 CC genotype that may be considered a protective factor [93]. Additionally, the (TAAAA)9 repeat and Asp alleles (Asp327Asn polymorphism) of the *SHBG* gene draw a decreasing trend of SHBG concentration, resulting in increased risk for male infertility [94].

#### 3.4.3. Hypogonadism and SHBG

Hypogonadism is described as a hypothalamic–pituitary–gonadal axis disruption of any kind. Treatment differs regarding the cause of hypogonadism and includes sex steroid replacement, GnRH or gonadotropin replacement, sometimes surgical interference or radiation therapy [95,96].

Hypogonadism in men and its connection to SHBG are being widely researched. According to the latest knowledge, many factors contribute to the onset of hypogonadism in men alongside aging [97,98]. Prolonging the age of 40 years, SHBG levels in men increase, and testosterone concentration decreases [97,98]. In contrast, the research conducted by Winters et al. finds a positive correlation between SHBG and total testosterone in untreated men with adult-onset hypogonadism [99].

Not only age, but other co-occurring diseases may contribute to alterations in SHBG concentration in men with hypogonadism. One of the most studied and verified comorbidities is human immunodeficiency virus (HIV) infection. Men living with HIV infection are considered at much higher risk of developing hypogonadism than healthy age-matched controls [100]. Considering this specific group of patients, total testosterone and free testosterone should be monitored with the use of liquid chromatography-tandem mass spectrometry due to the underestimation of the chemiluminescent immunoassay method [101]. In accordance with that, SHBG concentration should always be performed in HIV-positive males to calculate free testosterone concentration and not omit biochemical hypogonadism [101,102].

Furthermore, metabolic syndrome in men correlates with low levels of SHBG. Jarecki et al. found this association very promising in terms of the early diagnostic approach of metabolic syndrome in young men [103]. Moreover, the *SHBG* rs1799941 polymorphism in non-diabetic obese males is found to be a genetic risk factor for obesity-based hypogonadism [104]. Interestingly, chronic kidney disease (CKD) is a factor responsible for decreasing testosterone levels, depending on the CKD stage, whereas SHBG remains consistent [105].

Specifically, the research concerning women’s hypogonadism and SHBG is poorly studied. Nevertheless, interesting facts are known about how the SHBG concentration may relate to hypogonadism in females. Similarly to men, women undergoing hemodialysis are found to develop hypogonadism in the end-stage kidney disease. However, in contrast to males in this state, women’s SHBG concentration does not correlate with a high risk of death or cardiovascular event [106]. An interesting study involving Turner syndrome (TS) women was conducted to evaluate differences between TS women treated with hormone replacement therapy (HRT) and the non-treated group [107]. Overall, TS women are found to possess 30–50% lower SHBG concentration in comparison to healthy controls (non-TS), and no difference was found relating to the HRT usage. TS women with ongoing HRT had higher SHBG levels than TS non-HRT women and a decrease in androgen concentrations [107].

All relations mentioned in this subsection concerning SHBG are collected and summarized in Figure 5.

#### 3.4.4. Correlations Between Oral Contraception and SHBG

Nowadays, contraception is an essential element of appropriate sexual and reproductive healthcare [108]. In 2017–2019 in the United States, 65% of women aged 15–49 were using some kind of contraception. Around 16% of them chose the oral contraceptive pills as their contraception method [109]. OCs are found to increase SHBG levels, in some cases, even after discontinuing OC use, SHBG levels decreased but never returned to baseline references (before the first use of OC) [110].

Nowadays, the COCs are in favor of progestin-only pill (POPs) because of the strict dosing schedule of POPs, and many non-contraceptive benefits that come alongside taking COCs, such as menstrual regulation or acne and hirsutism improvement [111]. Nevertheless, estrogen-containing medications (including COCs) require detailed assessment of the patient in case of increased risk of venous thromboembolism [108,111]. Recently, the CDC updated the medical eligibility criteria for contraceptive use in 2024 [108].

COCs are responsible for increasing SHBG and lowering both total and free testosterone. Free testosterone decreases twice as much as total testosterone. Interestingly, the formula of COCs does not influence the decrease rate of testosterone, however, it does affect SHBG levels [110,112]. The biggest impact on increasing the SHBG is connected with higher doses of estrogen component and with the third and fourth generation of progestin. Second progestin generation and lower doses of ethinyl estradiol (20–25 µg) cause the smallest changes in the increase in SHBG [112]. In studies, authors successfully restored total testosterone levels in patients taking COCs by administering 50 mg of dehydroepiandrosterone a day, however, free testosterone was restored by 47% of baseline (before COC treatment) [113]. COC containing nomegestrol acetate and 17β-estradiol did not interfere with lipid parameters in comparison to other COCs (levonorgestrel and ethinylestradiol), however, it resulted in higher concentrations of SHBG [114]. What is more, the combination of estetrol and drospirenone effect on SHBG was less defined in comparison to COCs containing ethinylestradiol [115].

Oral contraceptives (OC) used in women with PCOS also elevate SHBG concentration. Especially, OCs containing ethinylestradiol and progestogens are responsible for the decrease in testosterone and androstenedione, and the increase in SHBG [116]. Elevated concentration of SHBG is particularly evident during the usage of OC consisting of ethinylestradiol and drospirenon or chlormadinone acetate. This combination is biochemically more effective due to the anti-androgenic effect of drospirenon and chlormadinone acetate [116].

Interestingly, discontinuation of taking OCs (ethinylestradiol + drospirenon) in patients with PCOS results in a decrease in SHBG concentration to base values after only 8 weeks [117]. Recently, a meta-analysis confirmed the better effect of ethinyloestradiol + cyproterone acetate on reducing hirsutism and biochemical hyperandrogenism in comparison to conventional OCs. However, due to higher risk of venous thrombotic events this combination is not recommended in PCOS patients as a first-line treatment [118].

What is more, SHBG marks its usefulness as a marker to assess the thrombotic risk of hormonal contraceptives in women [119]. Venous thromboembolism (VTE) consists of pulmonary embolism and deep vein thrombosis (mostly legs, splanchnic, arms, and cerebral veins) [120]. Risk factors for VTE are divided into strong and weak, and estrogen therapy is classified as a weak transient risk factor [120]. An interesting study by Tian et al. found a study to find possible relationship between hormones and VTE risk. The *SHBG* rs858518 is responsible for lowering the SHBG concentrations, which results in higher plasma levels of estradiol and increased risk of VTE in women [121].

### 3.5. SHBG and Thyroid Dysfunction

#### 3.5.1. Thyroid Hormones and SHBG

Thyroid function is closely related to sex hormone production. SHBG has become the interest of recent research. The SHBG concentration may be regulated indirectly by thyroid hormones’ effect on metabolism [28]. The link between SHBG and thyroid hormones is found in the liver and is known as HNF-4α. Thyroid hormones account for increased levels of HNF-4α, responsible for the promotion of the *SHBG* gene and its higher transcriptional activity, leading to increased production of SHBG [28]. Another study proves the effect of D-thyroxine on the increase in SHBG concentration. Interestingly, the dosage of 4 mg of D-thyroxine resulted in elevated SHBG levels. Women are found to be more affected by higher SHBG concentrations in comparison to men [122]. What is more, triiodothyronine (T3) significantly increases the SHBG levels as well [123,124]. One study compared the effect of regular tablet levothyroxine and liquid levothyroxine of ethanol-free formula in patients with hypothyroidism (both primary and central) [125]. Interestingly, in case of the liquid levothyroxine the SHBG concentration increased in both groups of hypothyroid patients. What is more, low-density lipoprotein (LDL) concentration decreased. This fact indicates a better response to the treatment and positive effects of the usage of liquid levothyroxine [125]. A recent study aimed to demonstrate the validity of twice-daily treatment with liothyronine and levothyroxine in patients after total thyroidectomy. According to results, no significant changes in SHBG were noticed in comparison to the control group (only levothyroxine in the morning) [126]. The Mendelian randomization study conducted by Kjaergaard et al. confirmed above mentioned correlation between thyroid hormones and sex hormones, including SHBG [127].

#### 3.5.2. SHBG and Hypothyroidism

As mentioned before, altered thyroid function, leading towards hypothyroidism, induces implications on SHBG concentrations [28,122,123,124,125,126,127]. Hypothyroid subjects are found to possess lower SHBG concentrations in comparison to healthy controls. In hypothyroidism, levothyroxine substitution leads to higher SHBG levels, normalization of total cholesterol and triglycerides, and an increase in HDL concentration [128]. What is more, this phenomenon marks its influence on lower SHBG concentrations followed by higher free and total testosterone levels in postmenopausal women with HT diagnosis, thus occurring autoimmunity in patients with hypothyroidism [129]. In men, the presence of anti-thyroid peroxidase antibodies (anti-TPO) leads to lower SHBG concentration, nevertheless, this correlation is not an independent predictor of autoimmunity [130]. Interestingly, studies are not unambiguous in the case of SHBG and hypothyroidism per se [131,132].

#### 3.5.3. SHBG and Hyperthyroidism

As in the case of hypothyroidism, hyperthyroidism leads to increased concentration of SHBG.

Interestingly, Ueshiba et al. found that in women with Graves’ disease (GD), despite the treatment with anti-thyroid drugs, normalization of SHBG was not achieved even after 6 months of therapy [133]. Nevertheless, SHBG levels were constantly decreasing during the clinical trial [134]. Relations between hyperthyroidism, SHBG, and ultrasonographic ovarian image have been found. Women with clinical hyperthyroidism (mostly GD, toxic nodular goiter, HT) possessed higher SHBG concentrations in comparison to the healthy age-matched control group.

What is more, the treatment with the use of thiamazole resulted in normalization of SHBG. Ovarian abnormalities found in ultrasonography with ongoing hyperthyroidism can normalize after the adjustment of anti-thyroid drug-based treatment [134].

#### 3.5.4. SHBG—Thyroid Nodules and Cancer

Nowadays, the incidence of thyroid nodules (TN) and thyroid cancer (TC) is increasing around the globe. TNs are found in approximately 25% of the general population, causing a major diagnostic and management challenge [135]. Recent studies indicate a higher prevalence of TNs in patients with altered SHBG concentration. Interestingly, the SPECT-China study revealed the correlation between lower SHBG levels and the presence of TNs in men (without any hormone replacement therapy). Furthermore, not only lower levels, but also the lower quartiles of SHBG indicate a higher risk of TNs in men [136]. Nevertheless, patients with differentiated thyroid cancer (DTC) diagnosis, followed by thyroidectomy, may be sought for a short period of levothyroxine withdrawal to induce an increase in TSH. Such conduct leads to a significant decrease in SHBG levels through the mechanism of induced hypothyroidism [137,138].

Due to the higher prevalence of thyroid cancer in women, the possibility of sex hormone interference is suspected. Rinaldi et al. found higher thyroid cancer risk in pre- and perimenopausal women with increased testosterone and androstenedione levels, however, these results did not support the thesis of a major role of sex hormones effect on carcinogenesis in the thyroid [139]. Another Mendelian randomization study found an association between higher SHBG and reduced risk of prostate cancer and breast cancer in men, however, no correlation regarding SHBG and TC was found [140].

#### 3.5.5. Resistance to Thyroid Hormone and SHBG

Thyroid hormone resistance syndrome (THR) refers to the reduced effect of thyroid hormones on target tissues. From the pathophysiological point of view, THR can be divided into generalized, selective pituitary, and selective peripheral resistance to thyroid hormones [141,142]. To the latest knowledge, THR is caused by mutations in the thyroid hormone receptor beta (*THRB*) gene locus 3p24.2 or in the thyroid hormone receptor alpha (*THRA*) gene locus 17q21.1, accordingly responsible for thyroid hormone resistance beta (THR-β) and thyroid hormone resistance alpha (THR-α) [143,144].

THR-β is inherited in a mostly autosomal dominant manner. Due to elevated thyroid hormone concentrations occurring with high or normal levels of TSH (non-suppressed), most likely lead to the suspicion of the THR-β diagnosis [16,145]. THR-α is inherited in an autosomal dominant matter or may be a result of sporadic mutations. Biochemically, THR-α differs from THR-β in thyroid function tests that, in most cases, reveal an extraordinarily low fT4/fT3 ratio [146]. The THR clinical presentation differs in patients. It may be associated with symptoms of hyperthyroidism, hypothyroidism, or patients may remain euthyroid [145].

Despite elevated levels of thyroid hormones, SHBG in THR remains within reference values, which is an important difference in comparison to hyperthyroid patients [144,147]. Nevertheless, according to Ueshiba et al. studies, there appears to be a delay in the normalization of SHBG following the initiation of anti-thyroid therapy in hyperthyroid patients. Therefore, six months after the commencement of the treatment, clinical assessment of the SHBG decreasing pattern should be undertaken, rather than solely reporting the value of SHBG [133]. Studies reveal a decrease in SHBG in most children with THR after the administration of T3, however, this phenomenon remains uncertain due to a small study group in the trial [148].

The above-mentioned issues related to SHBG and thyroid abnormalities are summarized in Figure 6.

### 3.6. SHBG and Cancers

#### 3.6.1. Breast Cancer

Breast cancer (BC) is a major worldwide problem. New analysis models concerning future epidemiology of BC suggest growth of BC occurrence to over 3 million new cases in 2040, followed by 1 million deaths [149]. Thus, global efforts in detecting potential risk factors in BC development must be made. It appears that SHBG and the *SHBG* gene have an impact on the occurrence of BC. Higher levels of SHBG are a protective factor against estrogen-positive receptor (ER+) BC formation and BCs overall; however, they may increase the occurrence of estrogen-negative receptor (ER−) BCs [150].

Variant C allele rs10454142 in *PPP1R21* results in lower SHBG levels, which increases the incidence of BCs in obese women [151,152]. Potentially, the rs6257 *SHBG* SNP may be a BC risk factor in post-menopausal women, however, larger-scale studies must be conducted to further evaluate this relationship [153]. Additionally, the *SHBG* Asp327Asn polymorphism was found to be a possible risk factor in postmenopausal Asian women [154]. What is more, a protective role of the D327N *SHBG* was found to play a role in women receiving hormone replacement therapy for menopause [155]. The link between SHBG and BC is complex and needs an understanding of many possible pathways that regulate the SHBG concentration in women. Recent data suggests other SHBG-connected polymorphisms which influence the BC occurrence, i.e., rs7910927 *JMJD1C*, rs780093 *GCKR*, rs440837 *ZBTB10,* and rs17496332 *PRMT6* [151].

#### 3.6.2. Ovarian Cancer

SHBG expression is present in ovarian cancer (OC) and is found to unfavorably impact the pathological features of the cancer [156]. Interestingly, the rs9898876 *SHBG* SNP decreases the risk of OC, however, no correlation between this specific SNP and progression of OC was found [157]. Hypothetically, SNP D356N (rs6259) in the *SHBG* gene may be associated with reduced risk of OC, nevertheless, more studies must be performed to confirm this susceptibility [158].

#### 3.6.3. Endometrial Cancer

Negative correlation between SHBG concentrations and the endometrial cancer (EC) risk was found [159]. The *SHBG* Asp327Asn polymorphism (rs6259) is a protective factor against the risk of EC, thus, it is responsible for higher values of SHBG [160,161]. What is more, the relationship between rs6259, dietary intake of polyphenol (soy isoflavones, tea consumption), and the risk of EC was confirmed [161].

#### 3.6.4. Lung Cancer

The connection between lung cancer and sex hormones overall seems non-intuitive, however, recent studies confirm the negative causal relationship between SHBG concentration and lung squamous cell carcinoma [162]. Interestingly, *UGT2B7* is a gene that regulates the SHBG levels. SNP rs12233719 is associated with the risk of non-small cell lung cancer in never-smoking women [163].

#### 3.6.5. Esophageal Cancer

*SHBG* SNPs are found to affect the risk of esophageal squamous cell carcinoma. Especially the C allele of rs727428 in *SHBG* is correlated with reduced risk of esophageal squamous cell carcinoma [164].

#### 3.6.6. Hepatocellular Carcinoma

Hepatocellular carcinoma (HCC) occurrence is modified by sex hormones. Higher SHBG levels act as risk factors for HCC in both males and females. The risk is 6 to 8 times higher for every doubled reference value [165]. The literature is not clear on whether the rs6259 of *SHBG* is a risk factor for HCC. One study found an association between Asp327Asn and a higher risk of HCC occurrence [166]. On the contrary, another study could not confirm this relationship [167].

All mentioned genetic aspects concerning SHBG and oncologic diseases are summarized in Table 3.

## 4. Discussion

SHBG is not just a transport protein, but has many different functions at the tissue, cellular, and molecular levels. This protein transports sex hormones but also appears to be an important link between endocrine, genetic, and metabolic processes. In this article, we have collected available studies on SHBG in various disease entities. All of the above-described entities have a certain relationship in common—the possibility of SHBG concentration influencing the course of the disease.

According to the studies presented earlier, SHBG synthesis is regulated by a number of factors closely related to carbohydrate metabolism, lipid metabolism, and the presence of pro-inflammatory cytokines produced by overdeveloped adipose tissue [10,27,28]. Excessive glucose intake and de novo lipogenesis processes result in inhibition of SHBG synthesis through effects on transcription factors [31]. Abnormalities in SHBG levels, caused by a number of metabolic abnormalities, result in abnormal hormone levels and direct clinical presentation, such as increased risk of developing hyperandrogenism in obese female populations or abnormal semen composition in men with overweight.

Due to the close correlation between metabolic disorders, an attempt was made to assess the usefulness of SHBG assessment as a risk marker for MS, DM2, and cardiovascular incidents.

Alterations in glucose serum levels and lipid metabolism in MS patients lead to reduced serum SHBG concentrations. Some of the statistical analyses confirmed the usefulness of SHBG serum levels as one of the markers of MS [34]. Also, decreased levels of SHBG were correlated with increased risk of DM2 development, especially in the male population. Higher concentration of SHBG in the female population seems to be a protective factor against DM2. Female patients with decreased SHBG frequently presented with IR rather than DM2 [63]. Interestingly, genetic variants of SHBG may play a crucial role in DM2 development. Patients with *SHBG* rs1799941 and rs6259 alleles presented a lower risk of developing DM2 in both male and female groups [72]. On the contrary, the rs6257 *SHBG* variant was associated with increased risk of DM2 due to the effect of lowering the concentration of SHBG [72].

The addition of SHBG to the panel of tests in MS or DM2 patients could be beneficial not only by assessing the risk of MS or DM2 itself, but also in combination with sex hormone assessment, allows for a more detailed evaluation of the endocrine system and the resolution of significant disorders from the reproductive system associated with metabolic disorders. However, more studies on larger groups of patients with metabolic disorders, diverse in terms of age and gender, should be performed to establish the exact usefulness of SHBG.

Based on previous studies, it was found that normal or increased SHBG levels were associated with a better metabolic profile of patients, proper glucose management, and consequently reduced cardiovascular risk. The improvement of SHBG levels in serum seems to be important, also due to the direct action of this protein in different tissues. Studies have shown that SHBG regulates the activity of myocardial fibroblasts, thereby reducing the risk of coronary heart disease and its remodeling [63].

The influence of SHBG on the activity of the cell mitochondria also seems to be important. The correct process of oxidation of organic compounds and oxidative phosphorylation reduces the amount of ROS produced [67]. The reduced amount of free radicals protects the vascular epithelium from damage, preventing the development of atherosclerosis, but also grants a protective effect on the genetic material of cells, preventing carcinogenesis.

Interestingly, studies have been conducted using adiponectin, which, after administration to HepG2 cells, caused an increase in SHBG levels [36]. In another study conducted by Liu et al., it has been proven that intake of β-carotene promotes SHBG and GLUT4 expression, indicating that β-carotene promotes glucose transport and inhibits IR in GDM by increasing the expression of SHBG [14]. These studies may serve as the basis for new methods of treating obesity and reducing the risk of its complications.

SHBG can also be used as an indicator of GDM. This is a type of diabetes that first appears during pregnancy. Its development is influenced by both hormonal factors and metabolic profile. Previous studies have examined the usefulness of SHBG as a marker of the risk of this type of diabetes [76,77,78]. Some of them confirm reduced levels of this protein in the group of patients who revealed GDM [15,76]. It is worth emphasizing that the increased risk of developing GDM also depends on factors occurring before pregnancy, such as PCOS, the course of which is also closely related to changes in SHBG levels.

The SHBG parameter is not a diagnostic marker mentioned in the Rotterdam criteria for diagnosing PCOS. Overall, lower SHBG is responsible for higher concentrations of androgens (and free androgen index) [82]. Thus, genetic abnormalities in *SHBG*, such as rs35785886, 727428, are responsible for lower SHBG and a higher risk of PCOS in women [94]. What is more, lower SHBG in women with PCOS and co-occurring IR is more likely to manifest obesity [107,109]. Women with PCOS benefit from the usage of OCs containing ethinylestradiol and progestogens—specifically drospirenone, chlormadinone acetate, or cyproterone acetate [116,117,118,119].

SHBG provides protective effects against anovulation-type infertility in women and manifests a negative correlation with BMI in men [13,91]. Male infertility is affected by *SHBG* alterations: rs6259 and rs727428 [93]. Hypogonadism in men is an important clinical state that warrants testing the SHBG levels, as biochemical hypogonadism is easy to miss in HIV-positive patients [101,102]. *SHBG* rs1799941 was found to be a genetic risk factor for obesity-based hypogonadism in men [104].

The determination of SHBG level in women with PCOS and infertility seems to be justified as this parameter directly affects the level of androgenization in PCOS, and normal SHBG levels are a protective factor in the case of infertility caused by anovulation. Early diagnosis and treatment of PCOS and associated metabolic disorders can also prevent complications in later pregnancy, such as GDM. Moreover, HIV-positive men are a special group of patients who can benefit from SHBG testing, as this parameter is important for the diagnosis of biochemical hypogonadism.

Thyroid function impacts the concentration of SHBG. In GD, women with hyperthyroidism are found to have higher SHBG concentrations, despite the anti-thyroid treatment [138]. What is more, ovarian abnormalities associated with SHBG alterations can normalize in women with clinical hyperthyroidism after the usage of thiamazole [140]. SHBG in THR syndrome is a parameter of special value, thus, it may be used to differentiate THR from hyperthyroidism of other entities [144,147]. What is more, the prevalence between the lower quartiles of SHBG and the risk of TNs in men was found [134]. Thyroid abnormalities—hypothyroidism, hyperthyroidism, TNs, and THR syndrome are proven to be affected by the SHBG.

Considering all the correlations described above, we see a value in performing SHBG level tests in patients with hyperthyroidism and THR, as an important factor for differentiating between these two conditions. Correlation between SHBG and TNs in men seems to be a promising finding. Currently, there is only one China-SPECT study published; however, information regarding not only the fact of reduced SHBG and TNs, but also the lower the SHBG quartile, the higher the risk of TNs in men, is an alarming phenomenon that requires deeper investigation. Nevertheless, we see value in determining SHBG in men with thyroid diseases or with a past/family medical history of thyroid nodules.

Solely genetic alterations regarding *SHBG* and other genes affecting SHBG concentration are, per se, important factors in various diseases. Oncological use of this parameter is mostly described in the case of breast cancer. *SHBG* rs6257 and Asp327Asn polymorphism are found as risk factors, whereas D327N *SHBG* was found to be a protective factor in women with ongoing hormone replacement therapy [153,154,155]. Genetic alterations of the *SHBG* gene are also correlated with a lower risk of ovarian cancer, endometrial cancer [159,162,163], or esophageal squamous cell carcinoma [164], whereas some alleles increase the risk of non-small lung cancer—*UGT2B7* [163]. Genetic alterations affect not only oncological diseases but vascular pathologies as well. The *SHBG* rs858518 acts as a risk factor for VTE in women [121]. What is more, longer alleles of the promoter of *SHBG* ((TAAAA)n polymorphism) result in a higher risk of early atherosclerosis in healthy women [62].

The genetic alterations described above are rare. In addition, studies that confirm individual correlations are limited to one and are not always based on a decent study group. More articles based on much larger study groups are needed to fully understand the impact of *SHBG* genetic alterations on the diseases mentioned above. Of the described diseases, breast cancer stands out in particular. The correlation with SHBG is well documented. Higher SHBG concentrations are a protective factor against estrogen-positive receptor (ER+) BC formation and BCs overall, nevertheless, the increase in the occurrence of estrogen-negative receptor (ER−) BCs is confirmed [150]. Therefore, testing for SHBG concentration in patients with breast cancer may be justified, as is potential testing for alterations in the SHBG gene.

## 5. Conclusions

Summarizing all the previously discussed disease entities, we find specific diseases in the course of which we consider SHBG testing to be clinically significant. The addition of SHBG to the panel of tests in MS or DM2 patients could be beneficial not only by assessing the risk of MS or DM2 itself, but also in combination with sex hormone assessment, allows for a more detailed evaluation of endocrine system and the resolution of significant disorders from the reproductive system associated with metabolic disorders. Tests of SHBG levels could also be beneficial in the group of patients with cardiac risk. Thyroid changes, such as hyperthyroidism, THR, and men with a family history of thyroid nodules, are the target group for testing the concentration of SHBG. Additionally, women with PCOS and infertility due to anovulation should have their SHBG levels tested. HIV-positive men are a suggested group for SHBG testing to assess possible biochemical hypogonadism. Additionally, women diagnosed with breast cancer and women with a family history of breast cancer should have their levels of SHBG tested due to the effect of this protein on the occurrence of ER(+) and ER(-) BCs. Nevertheless, due to a small amount of scientific proof, more studies indicating new potential usage of SHBG analysis must be performed to properly assess its clinical usefulness. Further large-scale prospective investigations are warranted to substantiate the role of SHBG as a routine clinical biomarker and to evaluate its potential incorporation into diagnostic panels for endocrine and metabolic disorders. Regardless, these scientific reports enable to focus of the research on the usefulness of SHBG determination in the above-mentioned disease entities.

## Figures and Tables

**Figure 1 biomedicines-13-01207-f001:**
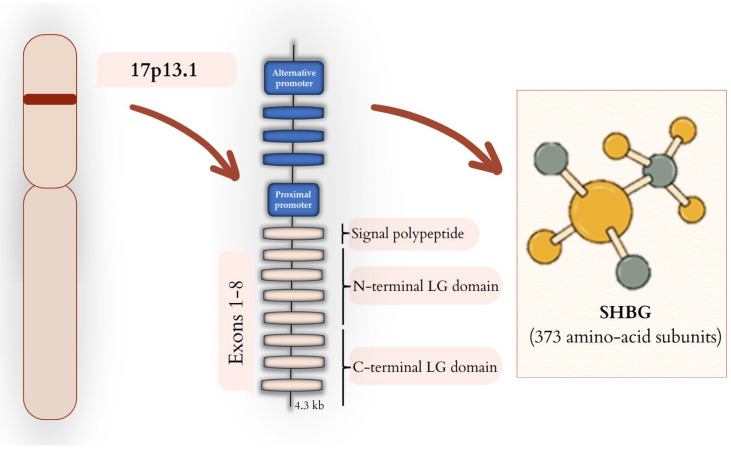
SHBG—from gene to protein. *SHBG* gene is located on the short arm of chromosome 17 (17p13.1). In the liver, SHBG expression is regulated by the proximal promoter that induces the transcription of mRNA composed of eight exons. The transcription process of individual exons leads to the synthesis of SHBG protein, which comprises a signal polypeptide, an N-terminal LG domain, and a C-terminal LG domain, which are built from 373 amino-acid subunits. SHBG expression in testis cells is dependent on an alternative promoter.

**Figure 2 biomedicines-13-01207-f002:**
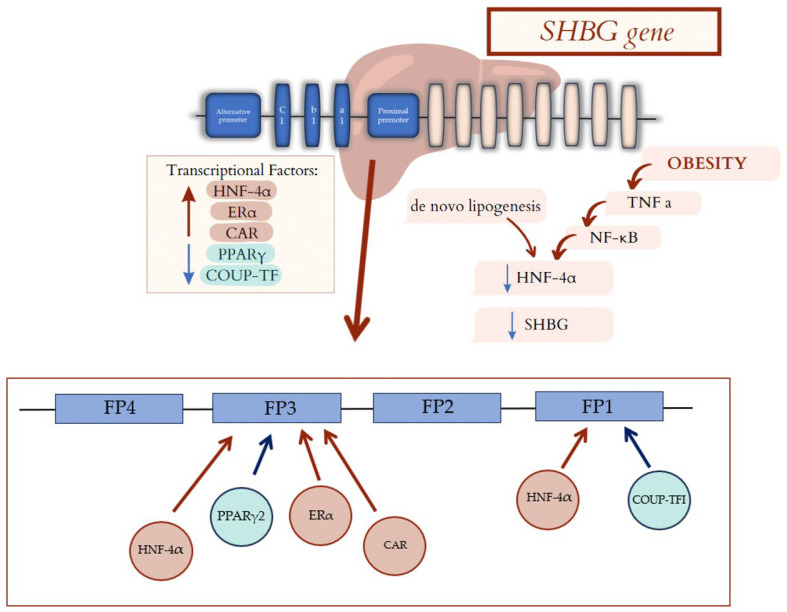
Factors affecting SHBG gene transcription. Transcriptional Factors: HNF-4α, Erα, and CAR cause an increased hepatic synthesis of the major SHBG fraction. PPARγ and COUP-TF lead to a decrease in gene expression and hepatic SHBG production. The gene for SHBG has a proximal promoter which has two trace regions (FP1 and FP3). FP1 bound COUP-TF and HNF-4α, which compete with each other for binding sites. FP3 bound HNF-4α, Erα, CAR, and PPARγ. Moreover, de novo lipogenesis and cytokine profile in obese patients decrease in HNF-4α, leading to decrease in SHBG. SHBG—sex hormone binding globulin; HNF-4α—hepatocyte nuclear factor 4 alpha; ERα—Estrogen Receptor α; CAR—constitutive androstane receptor; PPARγ—peroxisome proliferator-activated receptor gamma; COUP-TF—chicken ovalbumin upstream promoter transcription factor; TNF-α—tumor necrosis factor alpha; NF-κβ—nuclear factor-kappa B.

**Figure 3 biomedicines-13-01207-f003:**
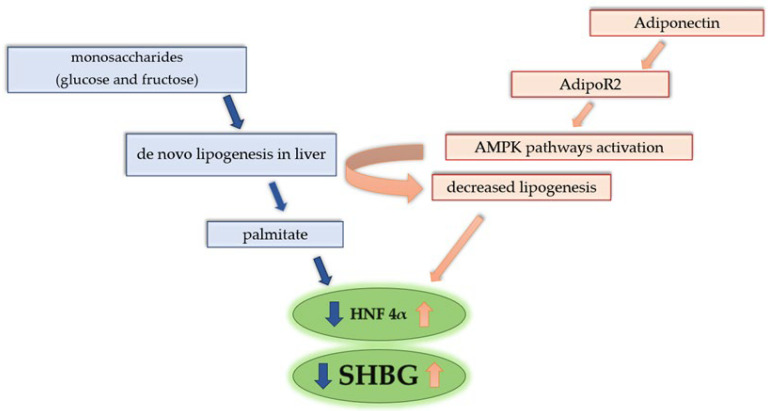
Correlation between de novo lipogenesis, adiponectin, and SHBG level. Increased intake of monosaccharides enhances de novo lipogenesis. Process of lipogenesis leads to an increased cellular concentration of palmitate, which results in downregulation of HNF-4α and indirectly decreased SHBG. Adiponectin acts through the activation of special receptors AdipoR2 in the liver, which results in AMPK pathways and influences the decreased process of de novo lipogenesis. Activation of AMPK pathways also increases oxidation of fatty acids in hepatic cells, resulting in decreased levels of lipids in hepatocytes. The influence of adiponectin on lipid metabolism increases HNF-4α, which enhances SHBG transcription.

**Figure 4 biomedicines-13-01207-f004:**
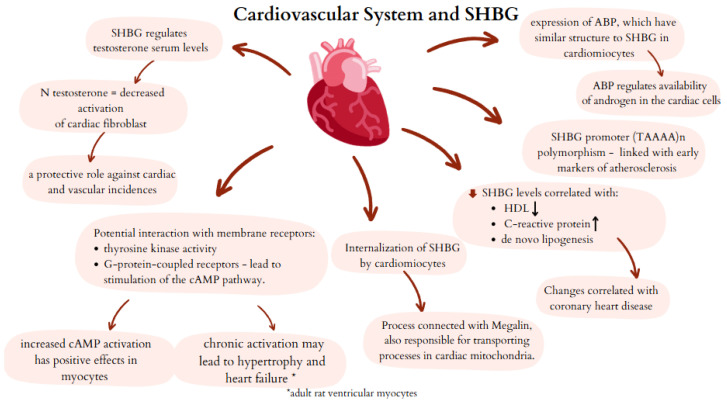
Potential mechanisms of SHBG activity in cardiovascular system. SHBG—sex hormone binding globulin; ABP—androgen binding protein; cAMP—cyclic adenosine monophosphate.

**Figure 5 biomedicines-13-01207-f005:**
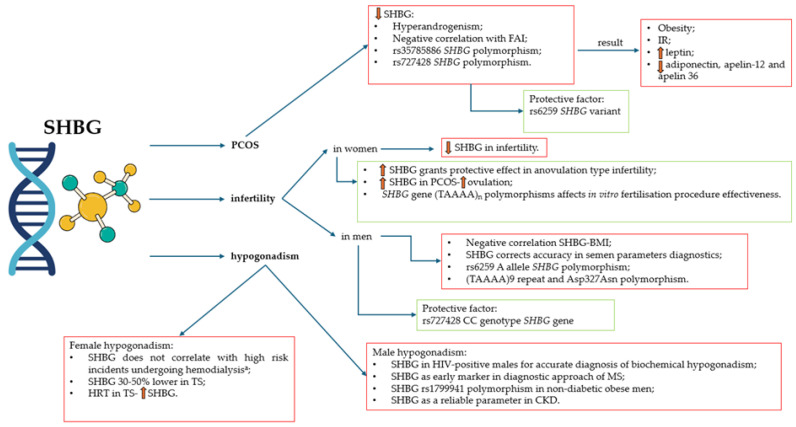
SHBG in PCOS, infertility, and hypogonadism. ^a^—high risk of death or cardiovascular event in end-stage kidney disease. SHBG—sex hormone-binding globulin; PCOS—polycystic ovary syndrome; FAI—free androgen index; IR—insulin resistance; BMI—body mass index; HIV—human immunodeficiency virus; MS—metabolic syndrome; CKD—chronic kidney disease; TS—Turner syndrome; HRT—hormone replacement therapy.

**Figure 6 biomedicines-13-01207-f006:**
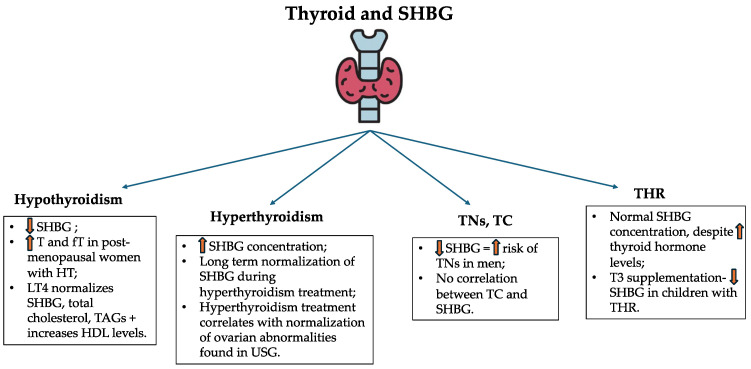
Serum SHBG concentrations in the aforementioned thyroid disorders. SHBG—sex hormone-binding globulin; T—testosterone; fT—free testosterone; HT—Hashimoto thyroiditis; LT4—levothyroxine; TAG—triglyceride; HDL—high-density lipoprotein; TN—thyroid nodule; TC—thyroid cancer; USG—ultrasonography; THR—thyroid hormone resistance; T3—triiodothyronine.

**Table 1 biomedicines-13-01207-t001:** Causes of SHBG serum level abnormalities [2,24].

SHBG Increase	SHBG Decrease
Weight loss	Overweight
Fasting	Obese
Growth hormone (pulsatility)	Lipogenesis
Thyroid hormones	Hepatic steatosis
Contraceptive pills	Cirrhosis
	Hypothyroidism

**Table 2 biomedicines-13-01207-t002:** Conditions correlate with SHBG.

Conditions Correlating with SHBG
Metabolic Syndrome
Cardiovascular Diseases
Diabetes Mellitus type 2
Polycystic Ovary Syndrome
Infertility
Hypogonadism
Oral Contraceptives
Hypothyroidism
Hyperthyroidism
Thyroid hormone resistance syndrome
Cancers

**Table 3 biomedicines-13-01207-t003:** Genetic alterations affecting SHBG levels. Note—both protective and risk factors are mentioned.

Cancer	Genetic Alteration
**Breast cancer**	*PPP1R21* rs10454142; *SHBG* rs6257; *SHBG* Asp^327^Asn polymorphism (rs6259); *JMJD1C* rs7910927; *GCKR* rs780093; *ZBTB10* rs440837; *PRMT6* rs17496332.
**Ovarian cancer**	*SHBG* rs9898876; *SHBG* rs6259.
**Endometrial cancer**	SHBG Asp327Asn polymorphism (rs6259);
**Lung cancer**	UGT2B7 rs12233719.
**Esophageal cancer**	SHBG rs727428.
**Hepatocellular carcinoma**	SHBG Asp327Asn polymorphism (rs6259).

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
