# Peer review of "New Insights in the Diagnostic Potential of Sex Hormone-Binding Globulin (SHBG)—Clinical Approach"

_biomedicines, 2025, doi:10.3390/biomedicines13051207_

Round 1
Reviewer 1 Report
Comments and Suggestions for Authors
1 The role of SHBG as an active signaling molecule is mentioned, including its interaction with G-protein-coupled receptors and activation of cAMP pathways. However, the manuscript does not clearly differentiate between experimental evidence and hypotheses, particularly when discussing chronic activation leading to hypertrophy. Specific citations for in vivo evidence would strengthen this section.
2. In the discussion of SHBG gene transcription regulation, the role of PPARγ and COUP-TF in inhibiting SHBG is well stated, but the competition mechanism with HNF-4α at FP1 and FP3 sites could be elaborated with a diagram showing binding site overlap and co-repressor recruitment.
3. The interpretation of adiponectin's role in SHBG regulation via HNF-4α is compelling, but the manuscript should mention that HepG2 is a hepatoma-derived cell line and may not fully replicate normal hepatocyte physiology. A note on the translational relevance of these findings would be helpful.
4. In discussing SHBG levels in PCOS, the text mentions that SHBG is not part of the Rotterdam criteria but then suggests its use in differentiating PCOS from functional hypothalamic amenorrhea. It would be valuable to suggest how SHBG could be integrated into current diagnostic workflows or scoring systems, rather than merely suggesting its utility.
5. The assertion that decreased SHBG contributes to hyperandrogenism is frequently repeated but not mechanistically broken down. Including a concise model showing the downstream effects of reduced SHBG on free androgen levels and their effects on ovarian theca cells or hypothalamic-pituitary signaling could clarify the pathophysiology.
6. The link between SHBG gene polymorphisms (e.g., rs6259, rs6257) and disease states is discussed extensively, but the text often fails to mention effect sizes or confidence intervals from the cited studies. Quantifying these associations would help assess clinical relevance.
7. The role of SHBG in cardiovascular risk assessment is discussed based on correlative data. However, causality is not established. The manuscript should clarify whether SHBG is a biomarker or a modifiable risk factor, particularly in light of studies like Li et al.'s meta-analysis.
8. In the thyroid function section, it is suggested that SHBG can be used to distinguish hyperthyroidism from thyroid hormone resistance. However, SHBG normalization lag after antithyroid therapy (as in Ueshiba et al.) complicates this. More critical discussion of the temporal dynamics of SHBG relative to thyroid function markers is needed.
9. Figure references (e.g., Figure 2, Figure 4, Figure 6) are repeatedly included in-text but without adequate descriptive captions or integration into the analysis. The manuscript should provide more interpretation and explanation of what each figure illustrates, especially for mechanistic pathways.
10. The mention of mitochondrial regulation by SHBG, based on Marycz et al.'s equine ASC model, introduces a novel perspective. However, the generalization to human clinical implications is premature without mentioning the species-specific limitations.
Reviewer 2 Report
Comments and Suggestions for Authors
Enclosed are my revisions in pdf file!

Round 2
Reviewer 1 Report
Comments and Suggestions for Authors
Authors made required corrections. Manuscript can be accepted with due approval of Editor.